# MARKUP-TO-IMAGE DIFFUSION MODELS WITH SCHEDULED SAMPLING

**Yuntian Deng[1], Noriyuki Kojima[2], Alexander M. Rush[2]**

[1] Harvard University `dengyuntian@seas.harvard.edu`
[2] Cornell University {`nk654,arush`}`@cornell.edu`

## ABSTRACT

Building on recent advances in image generation, we present a fully data-driven approach to rendering markup into images. The approach is based on diffusion models, which parameterize the distribution of data using a sequence of denoising operations on top of a Gaussian noise distribution. We view the diffusion denoising process as a sequential decision making process, and show that it exhibits compounding errors similar to exposure bias issues in imitation learning problems. To mitigate these issues, we adapt the scheduled sampling algorithm to diffusion training. We conduct experiments on four markup datasets: mathematical formulas (LaTeX), table layouts (HTML), sheet music (LilyPond), and molecular images (SMILES). These experiments each verify the effectiveness of the diffusion process and the use of scheduled sampling to fix generation issues. These results also show that the markup-to-image task presents a useful controlled compositional setting for diagnosing and analyzing generative image models.

## 1 INTRODUCTION

Recent years have witnessed rapid progress in text-to-image generation with the development and deployment of pretrained image/text encoders (Radford et al., 2021; Raffel et al., 2020) and powerful generative processes such as denoising diffusion probabilistic models (Sohl-Dickstein et al., 2015; Ho et al., 2020). Most existing image generation research focuses on generating realistic images conditioned on possibly ambiguous natural language (Nichol et al., 2021; Saharia et al., 2022; Ramesh et al., 2022). In this work, we instead study the task of markup-to-image generation, where the presentational markup describes exactly one-to-one what the final image should look like.

While the task of markup-to-image generation can be accomplished with standard renderers, we argue that this task has several nice properties for acting as a benchmark for evaluating and analyzing text-to-image generation models. First, the deterministic nature of the problem enables exposing and analyzing generation issues in a setting with known ground truth. Second, the compositional nature of markup language is nontrivial for neural models to capture, making it a challenging benchmark for relational properties. Finally, developing a model-based markup renderer enables interesting applications such as markup compilers that are resilient to typos, or even enable mixing natural and structured commands (Glennie, 1960; Teitelman, 1972).

We build a collection of markup-to-image datasets shown in Figure 1: mathematical formulas, table layouts, sheet music, and molecules (Nienhuys & Nieuwenhuizen, 2003; Weininger, 1988). These datasets can be used to assess the ability of generation models to produce coherent outputs in a structured environment. We then experiment with utilizing diffusion models, which represent the current state-of-the-art in conditional generation of realistic images, on these tasks.

The markup-to-image challenge exposes a new class of generation issues. For example, when generating formulas, current models generate perfectly formed output, but often generate duplicate or misplaced symbols (see Figure 2). This type of error is similar to the widely studied *exposure bias* issue in autoregressive text generation (Ranzato et al., 2015). To help the model fix this class of errors during the generation process, we propose to adapt scheduled sampling (Bengio et al., 2015).

**Math**

$$\widetilde{\gamma}_{\mathrm{hopf}} \simeq \sum_{n>0} \widetilde{G}_n \frac{(-a)^n}{2^{2n-1}}$$

```
\widetilde \gamma _ { \mathrm { h o p
f } } \simeq \sum _ { n > 0 } \widetilde {
G } _ { n } { \frac { ( - a ) ^ { n } } {
2 ^ { 2 n - 1 } } }
```

**Table Layouts**

```
...  f j
 </div> ...
```

**Sheet Music**

```
\relative c'' { \time 4/4 d4 |  r2 b4
b2 | ces4 b4˜ g2 f4 | a4 d8 | e4 g16 g2 f2
r4 | des2 d8 d8 f8 e4 d8 a16 b16 | d4 e2
d2. a8˜ g4 r16˜ e16. d2 f4 b4 e2 | f4. | b
16 a16 e4. r2˜ c4 r4 b4 d8 b2 | d4 | r8. e
8 e2 | r8˜ e2 }
```

**Molecules**

```
COc1ccc(cc1N)C(=O)Nc2ccccc2
```

Figure 1: Markup-to-Image suite with generated images. Tasks include mathematical formulas (LaTeX), table layouts (HTML), sheet music (LilyPond), and molecular images (SMILES). Each example is conditioned on a markup (bottom) and produces a rendered image (top). Evaluation directly compares the rendered image with the ground truth image.

Specifically, we train diffusion models by using the model's own generations as input such that the model learns to correct its own mistakes.

Experiments on all four datasets show that the proposed scheduled sampling approach improves the generation quality compared to baselines, and generates images of surprisingly good quality for these tasks. Models produce clearly recognizable images for all domains, and often do very well at representing the semantics of the task. Still, there is more to be done to ensure faithful and consistent generation in these difficult deterministic settings. All models, data, and code are publicly available at https://github.com/da03/markup2im.

## 2 MOTIVATION: DIFFUSION MODELS FOR MARKUP-TO-IMAGE GENERATION

**Task** We define the task of markup-to-image generation as converting a source in a markup language describing an image to that target image. The input is a sequence of $M$ tokens $x = x_1, \cdots, x_M \in \mathcal{X}$, and the target is an image $y \in \mathcal{Y} \subseteq \mathbb{R}^{H \times W}$ of height $H$ and width $W$ (for simplicity we only consider grayscale images here). The task of rendering is defined as a mapping $f : \mathcal{X} \to \mathcal{Y}$. Our goal is to approximate the rendering function using a model parameterized by $\theta$ $f_\theta : \mathcal{X} \to \mathcal{Y}$ trained on supervised examples $\{(x^i, y^i) : i \in \{1, 2, \cdots, N\}\}$. To make the task tangible, we show several examples of $x, y$ pairs in Figure 1.

**Challenge** The markup-to-image task contains several challenging properties that are not present in other image generation benchmarks. While the images are much simpler, they act more discretely than typical natural images. Layout mistakes by the model can lead to propagating errors throughout the image. For example, including an extra mathematical symbol can push everything one line further down. Some datasets also have long-term symbolic dependencies, which may be difficult for non-sequential models to handle, analogous to some of the challenges observed in non-autoregressive machine translation (Gu et al., 2018).

Figure 2: The generation process of diffusion (left) versus diffusion+schedule sampling (right). The numbers on the y-axis are the number of diffusion steps ($t$). The ground truth LaTeX is `\gamma_{n}^{\mu}=\alpha_{n}^{\mu}+\tilde{\alpha}_{n}^{\mu},~~~n\neq0`.

**Generation with Diffusion Models**   Denoising diffusion probabilistic models (DDPM) (Ho et al., 2020) parameterize a probabilistic distribution $P(y_0|x)$ as a Markov chain $P(y_{t-1}|y_t)$ with an initial distribution $P(y_T)$. These models conditionally generate an image by sampling iteratively from the following distribution (we omit the dependence on $x$ for simplicity):

$$P(y_T) = \mathcal{N}(0, I)$$
$$P(y_{t-1}|y_t) = \mathcal{N}(\mu_\theta(y_t, t); \sigma_t^2 I)$$

where $y_1, y_2, \cdots, y_T$ are latent variables of the same size as $y_0 \in \mathcal{Y}$, $\mu_\theta(\cdot, t)$ is a neural network parameterizing a map $\mathcal{Y} \to \mathcal{Y}$.

Diffusion models have proven to be effective for generating realistic images (Nichol et al., 2021; Saharia et al., 2022; Ramesh et al., 2022) and are more stable to train than alternative approaches for image generation such as Generative Adversarial Networks (Goodfellow et al., 2014). Diffusion models are surprisingly effective on the markup-to-image datasets as well. However, despite generating realistic images, they make major mistakes in the layout and positioning of the symbols. For an example of these mistakes see Figure 2 (left).

We attribute these mistakes to error propagation in the sequential Markov chain. Small mistakes early in the sampling process can lead to intermediate $y_t$ states that may have diverged significantly far from the model's observed distribution during training. This issue has been widely studied in the inverse RL and autoregressive token generation literature, where it is referred to as *exposure bias* (Ross et al., 2011; Ranzato et al., 2015).

## 3   SCHEDULED SAMPLING FOR DIFFUSION MODELS

In this work, we adapt scheduled sampling, a simple and effective method based on DAgger (Ross et al., 2011; Bengio et al., 2015) from discrete autoregressive models to the training procedure of diffusion models. The core idea is to replace the standard training procedure with a biased sampling approach that mimics the test-time model inference based on its own predictions. Before describing this approach, we first give a short background on training diffusion models.

**Background: Training Diffusion Models**   Diffusion models maximize an evidence lower bound (ELBO) on the above Markov chain. We introduce an auxiliary Markov chain $Q(y_1, \cdots, y_T|y_0) =$

$\prod_{t=1}^{T} Q(y_t|y_{t-1})$ to compute the ELBO:[1]

$$\log P(y_0) \geq \mathbb{E}_{y_1, \cdots, y_T \sim Q} \log \frac{P(y_0, \cdots, y_T)}{Q(y_1, \cdots, y_T)}$$

$$= \mathbb{E}_Q \left[ \log P(y_0|y_1) - \sum_{t=2}^{T} D_{\mathrm{KL}}(Q(y_{t-1}|y_t, y_0) \| P(y_{t-1}|y_t)) - D_{\mathrm{KL}}(Q(y_T|y_0) \| P(y_T)) \right]. \quad (1)$$

Diffusion models fix $Q$ to a predefined Markov chain:

$$Q(y_t|y_{t-1}) = \mathcal{N}(\sqrt{1 - \beta_t} y_{t-1}, \beta_t I) \quad Q(y_1, \cdots, y_T|y_0) = \prod_{t=1}^{T} Q(y_t|y_{t-1}),$$

where $\beta_1, \cdots, \beta_T$ is a sequence of predefined scalars controlling the variance schedule.

Since $Q$ is fixed, the last term $-\mathbb{E}_Q D_{\mathrm{KL}}(Q(y_T|y_0) \| P(y_T))$ in Equation (1) is a constant, and we only need to optimize

$$\mathbb{E}_Q \left[ \log P(y_0|y_1) - \sum_{t=2}^{T} D_{\mathrm{KL}}(Q(y_{t-1}|y_t, y_0) \| P(y_{t-1}|y_t)) \right]$$

$$= \mathbb{E}_{Q(y_1|y_0)} \log P(y_0|y_1) - \sum_{t=2}^{T} \mathbb{E}_{Q(y_t|y_0)} D_{\mathrm{KL}}(Q(y_{t-1}|y_t, y_0) \| P(y_{t-1}|y_t)).$$

With large $T$, sampling from $Q(y_t|y_0)$ can be made efficient since $Q(y_t|y_0)$ has an analytical form:

$$Q(y_t|y_0) = \int_{y_1, \cdots, y_{t-1}} Q(y_{1:t}|y_0) = \mathcal{N}(\sqrt{\bar{\alpha}_t} y_0, \sqrt{1 - \bar{\alpha}_t} I),$$

where $\bar{\alpha}_t = \prod_{s=1}^{t} \alpha_s$ and $\alpha_t = 1 - \beta_t$.

To simplify the $P(y_{t-1}|y_t)$ terms. Ho et al. (2020) parameterize this distribution by defining $\mu_\theta(y_t, t)$ through an auxiliary neural network $\epsilon_\theta(y_t, t)$:

$$\mu_\theta(y_t, t) = \frac{1}{\sqrt{\alpha_t}} (y_t - \frac{\beta_t}{\sqrt{1 - \bar{\alpha}_t}} \epsilon_\theta(y_t, t)).$$

With $P$ in this form, applying Gaussian identities, reparameterization (Kingma & Welling, 2013), and further simplification leads to a final MSE training objective,

$$\max_\theta \sum_{t=1}^{T} \mathbb{E}_{y_t \sim Q(y_t|y_0)} \left\| \frac{y_t - \sqrt{\bar{\alpha}_t} y_0}{\sqrt{1 - \bar{\alpha}_t}} - \epsilon_\theta(y_t, t) \right\|^2, \quad (2)$$

where $y_t$ is the sampled latent, $\bar{\alpha}_t$ is a constant derived from the variance schedule, $y_0$ is the training image, and $\epsilon_\theta$ is a neural network predicting the update to $y_t$ that leads to $y_{t-1}$.

**Scheduled Sampling**   Our main observation is that at training time, for each $t$, the objective function in Equation (2) takes the expectation with respect to a $Q(y_t|y_0)$. At test time the model instead uses the learned $P(y_t)$ leading to exposure bias issues like Figure 2.

Scheduled sampling (Bengio et al., 2015) suggests alternating sampling in training from the standard distribution and the model's own distribution based on a schedule that increases model usage through training. Ideally, we would sample from

$$P(y_t) = \int_{y_{t+1}, \cdots, y_T} P(t_T) \prod_{s=t+1}^{T} P(y_{s-1}|y_s).$$

---

[1]For a more detailed derivation, see Appendix B.

However, sampling from $P(y_t)$ is expensive since it requires rolling out the intermediate steps $y_T, \cdots, y_{t+1}$[2].

We propose an approximation instead. First we use $Q$ as an approximate posterior of an earlier step $t + m$, and then roll out a finite number of steps $m$ from $y_{t+m} \sim Q(y_{t+m}|y_0)$:

$$\tilde{P}(y_t|y_0) \triangleq \int_{y_{t+1}, \cdots, y_{t+m}} Q(y_{t+m}|y_0) \prod_{s=t+1}^{t+m} P(y_{s-1}|y_s).$$

Note that when $m = 0$, $\tilde{P}(y_t|y_0) = Q(y_t|y_0)$ and we recover normal diffusion training. When $m = T - t$, $\tilde{P}(y_t|y_0) = P(y_t)$ if $Q(y_T|y_0) = \mathcal{N}(0, I)$. An example of $m = 1$ is shown in Figure 3. Substituting back, the objective becomes

$$\sum_{t=1}^{T} \mathbb{E}_{y_t \sim \tilde{P}(y_t|y_0)} \left\| \frac{y_t - \sqrt{\bar{\alpha}_t} y_0}{\sqrt{1 - \bar{\alpha}_t}} - \epsilon_\theta(y_t, t) \right\|^2. \tag{3}$$

To compute its gradients, in theory we need to back-propagate through $\tilde{P}$ since it depends on $\theta$, but in practice to save memory we ignore $\frac{\partial \tilde{P}}{\partial \theta}$ and only consider the term inside expectation.

**Standard Diffusion**

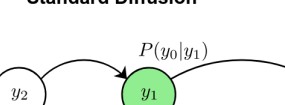

**Scheduled Sampling**

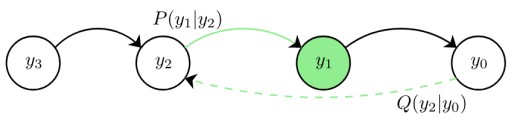

Figure 3: Diffusion samples $y_1$ from $Q$. Scheduled sampling instead samples an upstream latent variable $y_2$ and then $y_1$ based on the model's Markov chain $P(y_1|y_2)$.

## 4 MARKUP-TO-IMAGE SETUP

### 4.1 DATA

We adapt datasets from four domains to the task of markup-to-image. Table 1 provides a summary of dataset statistics.

**Math**  Our first dataset, LaTeX-to-Math, is a large collection of real-world mathematical expressions written in LaTeX markups and their rendered images. We adopt IM2LATEX-100K introduced in Deng et al. (2016), which is collected from Physics papers on arXiv. IM2LATEX-100K is originally created for the visual markup decompiling task, but we adapt this dataset for the reverse task of markup-to-image. We pad all images to size $64 \times 320$ and remove images larger than that size. For faster evaluation, we form a smaller test set by subsampling 1,024 examples from the original test set in IM2LATEX-100K .

**Table Layouts**  The second dataset we use is based on the 100k synthesized HTML snippets and corresponding rendered webpage images from Deng et al. (2016). Each HTML snippet contains a nested <div> with a solid border, a random width, and a random float. The maximum depth of a nest is limited to two. We make no change to this dataset, except that we subsample 1,024 examples from the original test set to form a new test set.

**Sheet Music**  We generate a third dataset of sheet music. The markup language LilyPond is a file format for music engraving (Nienhuys & Nieuwenhuizen, 2003). LilyPond is a powerful language for writing music scores: it allows specifying notes using letters and note durations using numbers. One challenge in the LilyPond-to-Sheet music task is to deal with the possible "relative" mode, where the determination of each note relies on where the previous note is. We generate 35k synthetic LilyPond files and compile them into sheet music. We downsample images by a factor of two and then filter out images greater than $192 \times 448$.

---

[2]There is no analytical solution since the transition probabilities in this Markov chain are parameterized by a neural network $\mu_\theta$.

| Dataset | Input Format | Input Length | # Train | # Val | # Test | Image Size | Grayscale |
|---|---|---|---|---|---|---|---|
| Math | LaTeX Math | 113 | 55,033 | 6,072 | 1,024 | $64 \times 320$ | Y |
| Table Layouts | HTML Snippet | 481 | 80,000 | 10,000 | 1,024 | $64 \times 64$ | Y |
| Sheet Music | LilyPond File | 240 | 30,902 | 989 | 988 | $192 \times 448$ | Y |
| Molecules | SMILES String | 30 | 17,925 | 1,000 | 1,000 | $128 \times 128$ | N |

Table 1: Markup-to-image datasets. Inputs to each dataset are described in Section 4.1 in detail. Input length is measured as the median number of characters in the validation set.

**Molecules**   The last dataset we use is from the chemistry domain. The input is a string of Simplified Molecular Input Line Entry System (SMILES) which specifies atoms and bonds of a molecule (Weininger, 1988). The output is a scheme of the input molecule. We use a solubility dataset by Wilkinson et al. (2022), containing 19,925 SMILES strings. The dataset is originally proposed to improve the accessibility of chemical structures for deep learning research. 2D molecules images are rendered from SMILES strings using the Python package RDKIT (Landrum et al., 2016). We partition the data into training, validation, and test sets. We downsample images by a factor of two.

## 4.2   EVALUATION

Popular metrics for conditional image generation such as Inception Score (Salimans et al., 2016) or Fréchet Inception Distance (Heusel et al., 2017) evaluate the fidelity and high-level semantics of generated images. In markup-to-image tasks, we instead emphasize the pixel-level similarity between generated and ground truth images because input markups describe exactly what the image should look like.

**Pixel Metrics**   Our primary evaluation metric is Dynamic Time Warping (DTW) (Müller, 2007), which calculates the pixel-level similarities of images by treating them as a column time-series. We preprocess images by binarizing them. We treat binarized images as time-series by viewing each image as a sequence of column feature vectors. We evaluate the similarity of generated and ground truth images by calculating the cost of alignment between the two time-series using DTW[3]. We use Euclidean distance as a feature matching metric. We allow minor perturbations of generated images by allowing up to 10% of upward/downward movement during feature matching.

Our secondary evaluation metric is the root squared mean error (RMSE) of pixels between generated and ground truth images. We convert all images to grayscale before calculating RMSE. While RMSE compares two images at the pixel level, one drawback is that RMSE heavily penalizes the score of symbolically equivalent images with minor perturbations.

**Complimentary Metrics**   Complementary to the above two main metrics, we report one learned and six classical image similarity metrics. We use CLIP score (Radford et al., 2021) as a learned metric to calculate the similarity between the CLIP embeddings of generated and ground truth images. While CLIP score is robust to minor perturbations of images, it is unclear if CLIP embeddings capture the symbolic meanings of the images in the domains of rendered markups. For classical image similarity metrics[4], we report SSIM (Wang et al., 2004), PSNR (Wang et al., 2004), UQI (Wang & Bovik, 2002), ERGAS (Wald, 2000), SCC (Zhou et al., 1998), and RASE (González-Audícana et al., 2004).

## 4.3   EXPERIMENTAL SETUP

**Model**   For Math, Table Layouts, and Sheet Music datasets, we use GPT-Neo-175M (Black et al., 2021; Gao et al., 2020) as the input encoder, which incorporates source code in its pre-training. For the Molecules dataset, we use ChemBERTa-77M-MLM from DeepChem (Ramsundar et al., 2019; Chithrananda et al., 2020) to encode the input. To parameterize the diffusion decoder, we experiment with three variants of U-Net (Ronneberger et al., 2015): 1) a standard U-Net conditioned on an

---

[3]We use the DTW implementation by https://tslearn.readthedocs.io/en/stable/user_guide/dtw.html.

[4]We use the similarity metric implementation by https://github.com/andrewekhalel/sewar.

| Approach | Pixel | | Complimentary | | | | | | |
|---|---|---|---|---|---|---|---|---|---|
| | DTW↓ | RMSE↓ | CLIP↑ | SSIM↑ | PSNR↑ | UQI↑ | ERGAS↓ | SCC↑ | RASE↓ |
| **Math** | | | | | | | | | |
| Stable Diffusion | 115.77 | 144.65 | 0.71 | 0.15 | 5.12 | 0.54 | 35366.43 | 0.00 | 8721.85 |
| Base-Attn,-Pos | 27.73 | 44.72 | 0.95 | 0.70 | 15.35 | 0.97 | 2916.76 | 0.02 | 729.19 |
| Base+Attn,-Pos | 20.81 | 39.53 | 0.96 | 0.76 | 16.62 | 0.98 | 2448.35 | 0.06 | 612.09 |
| Base+Attn,+Pos | 19.45 | 37.81 | 0.97 | 0.78 | 17.12 | 0.98 | 2314.31 | 0.07 | 578.58 |
| Scheduled Sampling | **18.81** | **37.19** | **0.97** | **0.79** | **17.25** | **0.98** | **2247.41** | **0.07** | **561.85** |
| **Table Layouts** | | | | | | | | | |
| Base+Attn,-Pos | 6.09 | 22.89 | 0.95 | 0.92 | 38.55 | 0.98 | 2497.51 | 0.44 | 624.38 |
| Base+Attn,+Pos | 5.91 | 22.17 | 0.95 | 0.93 | 38.91 | 0.98 | 2409.28 | 0.44 | 602.32 |
| Scheduled Sampling | **5.64** | **21.11** | **0.95** | **0.93** | **40.20** | **0.98** | **2285.83** | **0.45** | **571.46** |
| **Sheet Music** | | | | | | | | | |
| Base+Attn,-Pos | 81.21 | 45.23 | **0.97** | 0.67 | 15.10 | 0.97 | 3056.72 | **0.02** | 764.18 |
| Base+Attn,+Pos | 80.63 | 45.16 | 0.97 | 0.68 | 15.11 | 0.97 | 3032.40 | 0.02 | 758.10 |
| Scheduled Sampling | **79.76** | **44.70** | 0.97 | **0.68** | **15.20** | 0.97 | **2978.36** | 0.02 | **744.59** |
| **Molecules** | | | | | | | | | |
| Base+Attn,-Pos | 24.87 | 38.12 | **0.97** | 0.61 | 16.66 | 0.98 | 2482.08 | **0.00** | 620.52 |
| Base+Attn,+Pos | 24.95 | 38.15 | 0.96 | 0.61 | 16.64 | 0.98 | **2455.18** | 0.00 | **613.79** |
| Scheduled Sampling | **24.80** | **37.92** | 0.96 | **0.61** | **16.69** | **0.98** | 2467.16 | 0.00 | 616.79 |

Table 2: Evaluation results of markup-to-image generation across four datasets. (+/-)Attn indicates a model with or without attention, and (+/-)Pos is a model with or without positional embeddings. Scheduled Sampling is applied to training of models with attention and positional embeddings. Best results are in bold (based on full precision although the numbers are rounded to two decimal places).

average-pooled encoder embedding (denoted as "-Attn,-Pos"), 2) a U-Net alternating with cross-attention layers over the full resolution of the encoder embeddings (denoted as "+Attn,-Pos"), and 3) a U-Net with both cross-attention and additional position embeddings on the query marking row ids and column ids (denoted as "+Attn,+Pos") (Vaswani et al., 2017).

**Hyperparameters**  We train all models for 100 epochs using the AdamW optimizer (Kingma & Ba, 2014; Loshchilov & Hutter, 2018). The learning rate is set to $1e-4$ with a cosine decay schedule over 100 epochs and 500 warmup steps. We use a batch size of 16 for all models. For scheduled sampling, we use $m = 1$. We linearly increase the rate of applying scheduled sampling from 0% to 50% from the beginning of the training to the end.

**Implementation Details**  Our code is built on top of the HuggingFace diffusers library[5]. We use a single Nvidia A100 GPU to train on the Math, Table Layouts, and Molecules datasets; We use four A100s to train on the Sheet Music dataset. Training takes approximately 25 minutes per epoch for Math and Table Layouts, 30 minutes for Sheet Music, and 15 minutes for Molecules. Although one potential concern is that the scheduled sampling approach needs more compute due to the extra computation to get $\tilde{P}$ for $m > 0$, in practice, we find that the training speed is not much affected: on the Math dataset, scheduled sampling takes 24 minutes 59 seconds per training epoch, whereas without scheduled sampling it takes 24 minutes 13 seconds per epoch.

## 5 RESULTS

Table 2 summarizes the results of markup-to-image tasks across four domains. We use DTW and RMSE as our primary evaluation metrics to make our experimental conclusions. First, we train and evaluate the variations of diffusion models on the Math dataset. Comparing the model with attention ("-Attn,-Pos") to without attention ("+Attn,-Pos"), using attention in the model results in a

---

[5]https://github.com/huggingface/diffusers

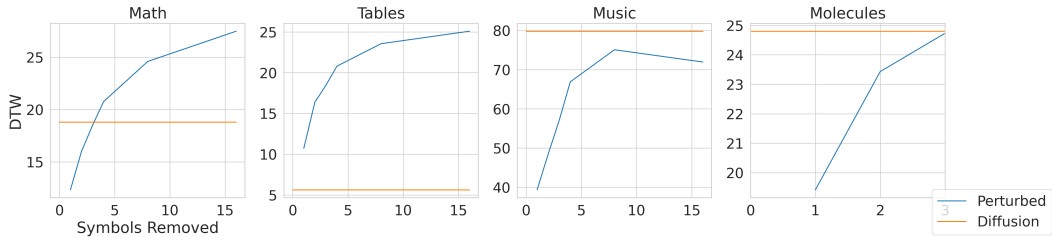

Figure 4: Perturbation results. For each dataset, we compare the DTW score of images generated by a compiler, removing an increasing number of symbols from markups, and our best model.

$$\Omega^{2(1+)} = -\frac{4Q}{\mu(1+q)} \frac{1}{P(I+cc_1)^2}.$$

$$\Omega^{2(1+q)} = -\frac{4Q}{\mu(1+q)} \frac{1}{P(I+c_1)^2}.$$

Figure 5: Qualitative results showing typical mistakes. (Top row) Model-generated images across datasets. (Bottom row) Ground truth images.

significant improvement by reducing DTW (25% reduction) and RMSE (12% reduction). Therefore, we always use attention for experiments on other datasets. We observe that additionally, using positional embeddings ("+Attn,+Pos") is helpful for the Math dataset. The proposed scheduled sampling approach improves the model's performance using attention and positional embeddings. We also evaluate pre-trained Stable Diffusion (Rombach et al., 2022), but our evaluation results suggest that pre-trained Stable Diffusion does not have a zero-shot transferability to the markup domain. We provide qualitative results for Stable Diffusion in Figure 14.

We observe a similar trend in the other three datasets–Table Layouts, Sheet Music, and Molecules. Using positional embeddings improves the performance measured by DTW and RMSE (except for the Molecules dataset). Training models with the proposed scheduled sampling achieves the best results consistently across all the datasets. As noted in Figure 2, we can qualitatively observe that schedule sampling, which exposes the model to its own generations during training time, comes with the benefits of the model being capable of correcting its own mistakes at inference time.

**Absolute Evaluation**    Our evaluation metrics enable relative comparisons between models in the markup-to-image task. However, it remains unclear how capable the models are in an absolute sense–if the models are generating near-perfect images or even the best model is missing a lot of symbols. We investigate this question by removing an increasing number of symbols from the ground truth markups and evaluating the perturbed images against the ground truth images. Our results in Figure 4 highlight that our best model performs roughly equivalent to the ground truth images with three symbols removed on the Math dataset. On the other hand, our best model performs better than ground truth images with only a single symbol removed on the Table Layouts dataset and two symbols removed on the Molecules dataset, indicating that our best model adapts to these datasets well. Results for music are less strong.

**Qualitative Analysis**    We perform qualitative analysis on the results of our best models, and we observe that diffusion models show a different level of adaptation to four datasets. First, we observe that diffusion models fully learn the Table Layouts dataset, where the majority of generated images are equivalent to the ground truth images for human eyes. Second, diffusion models perform moderately well on the Math and Molecules datasets: diffusion models generate images similar to the ground truth images most of the time on the Math dataset, but less frequently so on the Molecules dataset. The common failure modes such as dropping a few symbols, adding extra symbols, and repeating symbols are illustrated in Figure 5.

On the Sheet Music dataset, diffusion models struggle by generating images that deviate significantly from the ground truth images. Despite this, we observe that diffusion models manage to generate the first few symbols correctly in most cases. The intrinsic difficulty of the Sheet Music dataset is a long chain of dependency of symbols from left to right, and the limited number of denoising steps might be a bottleneck to generating images containing this long chain. We provide additional qualitative results for all four datasets in Appendix A.

## 6 RELATED WORK

**Text-to-Image Generation**   Text-to-image generation has been broadly studied in the machine learning literature, and several model families have been adopted to approach the task. Generative Adversarial Networks (Goodfellow et al., 2014) is one of the popular choices to generate realistic images from text prompts. Initiating from the pioneering work by Reed et al. (2016a), numerous approaches are developed to improve the quality of text-to-image generation (Reed et al., 2016b; Zhang et al., 2017; 2018; Zhu et al., 2019; Tao et al., 2020; Koh et al., 2021, inter alia). Another common method is based on VQ-VAE (Van Den Oord et al., 2017), treating text-to-image generation as a sequence-to-sequence task of predicting discretized image tokens autoregressively from text prompts (Ramesh et al., 2021; Ding et al., 2021; Gafni et al., 2022; Gu et al., 2022; Aghajanyan et al., 2022; Yu et al., 2022). Diffusion models (Sohl-Dickstein et al., 2015) are the most recent progress in text-to-image generation. The simplicity of training diffusion models introduces significant utility, which often reduces to the minimization of mean-squared error for estimating noises added to images (Ho et al., 2020). Diffusion models are free from training instability or model collapses (Brock et al., 2018; Dhariwal & Nichol, 2021), and yet manage to outperform Generative Adversarial Networks on text-to-image generation in the MSCOCO domain (Dhariwal & Nichol, 2021). Diffusion models trained on large-scale image-text pairs demonstrate impressive performance in generating creative natural or artistic images (Nichol et al., 2021; Ramesh et al., 2022; Saharia et al., 2022).

So far, the demonstration of successful text-to-image generation models is centered around the scenario with flexible interpretations of text prompts (e.g., artistic image generation). When there is an exact interpretation of the given text prompt (e.g., markup-to-image generation), text-to-image generation models are understudied (with a few exceptions such as Liu et al. (2021) which studied controlled text-to-image generation in CLEVR (Johnson et al., 2017) and iGibson (Shen et al., 2021) domains). Prior work reports that state-of-the-art diffusion models face challenges in the exact interpretation scenario. For example, Ramesh et al. (2022) report unCLIP struggles to generate coherent texts based on images. In this work, we propose a controlled compositional testbed for the exact interpretation scenario across four domains. Our study brings potential opportunities for evaluating the ability of generation models to produce coherent outputs in a structured environment, and highlights open challenges of deploying diffusion models in the exact interpretation scenario.

**Scheduled Sampling**   In sequential prediction tasks, the mismatch between teacher forcing training and inference is known as an exposure bias problem (Ranzato et al., 2015; Spencer et al., 2021). During teacher forcing training, a model's next-step prediction is based on previous steps from the ground truth sequence. During inference, the model performs the next step based on its own previous predictions. Training algorithms such as DAgger (Ross et al., 2011) or scheduled sampling (Bengio et al., 2015) are developed to mitigate this mismatch problem, primarily by forcing the model to use its own previous predictions during training with some probability. In this work, we observe a problem similar to exposure bias in diffusion models, and we demonstrate that training diffusion models using scheduled sampling improves their performance on markup-to-image generation.

## 7 CONCLUSION

We propose the task of markup-to-image generation which differs from natural image generation in that there are ground truth images and deterministic compositionality. We adapt four instances of this task to analyze state-of-the-art diffusion-based image generation models. Motivated by the observation that a diffusion model cannot correct its own mistakes at inference time, we propose to use scheduled sampling to expose it to its own generations during training. Experiments confirm our approach's effectiveness, although perfect rendering is not yet achieved. We believe that rendering markup is an interesting benchmark and a potential application of pretrained models with diffusion.

ACKNOWLEDGMENTS

YD is supported by an Nvidia Fellowship. NK is supported by a Masason Fellowship. AR is supported by NSF CAREER 2037519, NSF 1704834, and a Sloan Fellowship. Thanks to Bing Yan for preparing molecule data and Ge Gao for editing drafts of this paper. We would also like to thank Harvard University FAS Research Computing for providing computational resources.

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

# A    QUALITATIVE RESULTS

We provide additional qualitative results from models trained with or without scheduled sampling on four datasets in Figure 6, Figure 7, Figure 8, and Figure 9.

$$a = \frac{\beta^2}{4\pi} \left( \frac{1}{1 + \beta^2/4\pi} \right) \quad . \qquad a = \frac{\beta^2}{4\pi} \left( \frac{1}{1 + \beta^{2/2}/4} \right) \quad . \qquad a = \frac{\beta^2}{4\pi} \left( \frac{1}{1 + \beta/4\pi} \right) \quad .$$

$$c_{ijk} = \frac{\partial^3 F}{\partial t^i \, \partial t^j \, \partial t^k} \qquad\qquad c_{ijk} = \frac{\partial^3 F}{\partial t^i \, \partial^i \, \partial^k} \qquad\qquad c_{ijk} = \frac{\partial^3 F}{\partial t^i \partial^{jj} \, \partial t'^k}$$

$$\langle f \rangle \equiv \int D\xi \, f \, P[\xi, \{\phi\}] \qquad \langle f \rangle \equiv \int D \, \xi \; P[\xi, \{\{\phi\} \qquad \langle f \rangle \equiv \int D\xi \, f \, P[\xi, \{\phi\}]$$

$$A(\varsigma) = \int du \; a(u) e^{iu \cdot \varsigma}, \qquad A(\varsigma) = \int du \; a(\imath) \, e^{iu \cdot \varsigma}, \qquad A(\varsigma) = \int du \; a(u) e^{iu \cdot \varsigma},$$

Figure 6: Qualitative results in the Math domain. Left column: ground truth images. Middle column: generations from +Attn,+Pos. Right column: generations from Scheduled Sampling. The top two rows are random selections, and the bottom two rows are examples of good generations.

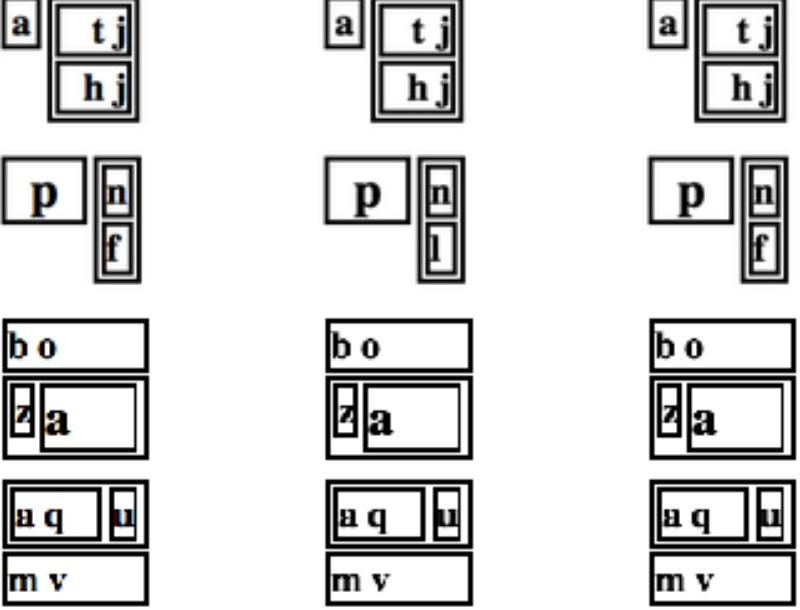

Figure 7: Qualitative results in the Table Layouts domain. Left column: ground truth images. Middle column: generations from +Attn,+Pos. Right column: generations from Scheduled Sampling. The top two rows are random selections, and the bottom two rows are examples of good generations.

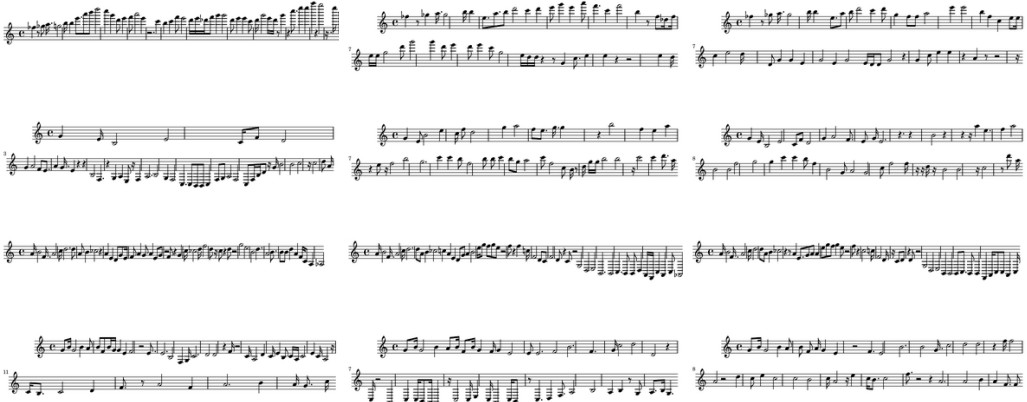

Figure 8: Qualitative results in the Sheet Music domain. Left column: ground truth images. Middle column: generations from +Attn,+Pos. Right column: generations from Scheduled Sampling. The top two rows are random selections, and the bottom two rows are examples of good generations.

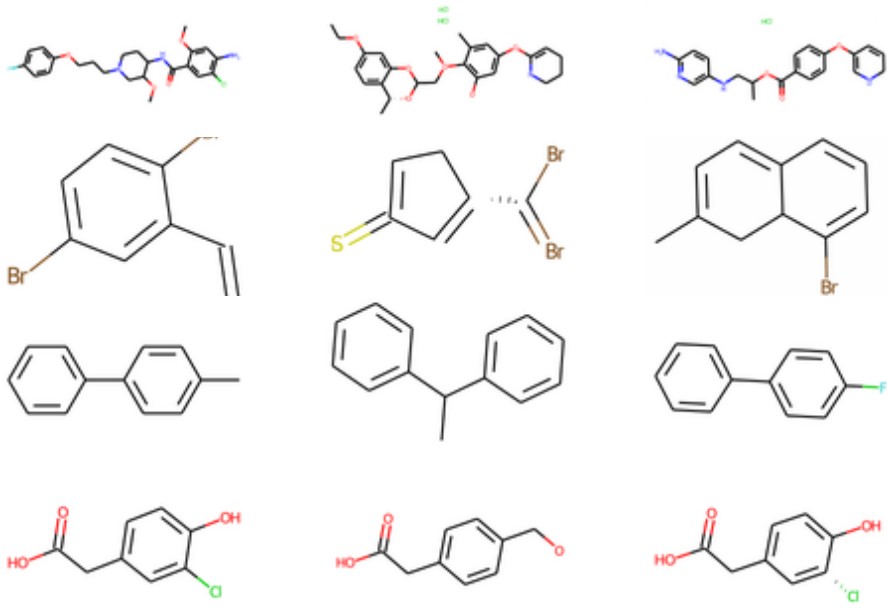

Figure 9: Qualitative results in the Molecules domain. Left column: ground truth images. Middle column: generations from +Attn,+Pos. Right column: generations from Scheduled Sampling. The top two rows are random selections, and the bottom two rows are examples of good generations.

## B  DETAILED DERIVATION OF THE DIFFUSION OBJECTIVE

In this section, we provide a more detailed derivation of the objective function of the diffusion process.

**Generative Process**  Denoising diffusion probabilistic models (DDPM) (Ho et al., 2020) parameterize the probabilistic distribution $P(y_0|x)$ by parameterizing a Markov chain $P(y_{t-1}|y_t)$ with an initial distribution $P(y_T)$ (we omit the dependence on $x$ for simplicity):

$$P(y_T) = \mathcal{N}(0, I)$$
$$P(y_{t-1}|y_t) = \mathcal{N}(\mu_\theta(y_t, t); \sigma_t^2 I)$$
$$P(y_{0:T}) = P(y_T) \prod_{t=1}^{T} P(y_{t-1}|y_t),$$

where $y_1, y_2, \cdots, y_T$ are latent variables of the same size as $y_0 \in \mathcal{Y}$, $\mu_\theta(\cdot, t)$ is a neural network parameterizing a map $\mathcal{Y} \to \mathcal{Y}$.

**Evidence Lower Bound**  To learn the model, we use variational inference and maximize the evidence lower bound (ELBO) (Bishop & Nasrabadi, 2006). We introduce an auxiliary Markov chain $Q(y_1, \cdots, y_T|y_0) = \prod_{t=1}^{T} Q(y_t|y_{t-1})$ as the approximate posterior distribution to compute the ELBO:

$$\log P(y_0) \geq \mathbb{E}_{y_1, \cdots, y_T \sim Q} \log \frac{P(y_0, \cdots, y_T)}{Q(y_1, \cdots, y_T)}$$

$$= \mathbb{E}_Q \log \frac{P(y_T) \prod_{t=1}^{T} P(y_{t-1}|y_t)}{Q(y_1|y_0) \prod_{t=2}^{T} Q(y_t|y_{t-1})}$$

$$= \mathbb{E}_Q \log \frac{P(y_T) \prod_{t=1}^{T} P(y_{t-1}|y_t)}{Q(y_1|y_0) \prod_{t=2}^{T} [Q(y_{t-1}|y_t, y_0)Q(y_t|y_0)/Q(y_{t-1}|y_0)]}$$

$$= \mathbb{E}_Q \log \frac{P(y_T)P(y_0|y_1) \prod_{t=2}^{T} P(y_{t-1}|y_t)}{Q(y_T|y_0) \prod_{t=2}^{T} Q(y_{t-1}|y_t, y_0)}$$

$$= \mathbb{E}_Q \left[ \log P(y_0|y_1) + \log \prod_{t=2}^{T} \frac{P(y_{t-1}|y_t)}{Q(y_{t-1}|y_t, y_0))} + \log \frac{P(y_T)}{Q(y_T|y_0)} \right]$$

$$= \mathbb{E}_Q \left[ \log P(y_0|y_1) - \sum_{t=2}^{T} D_{\mathrm{KL}}(Q(y_{t-1}|y_t, y_0) \| P(y_{t-1}|y_t)) - D_{\mathrm{KL}}(Q(y_T|y_0) \| P(y_T)) \right]. \quad (4)$$

Note that this shows how Equation (1) in Section 3 in the main paper is derived.

Instead of optimizing $Q$ to maximize ELBO as is typically done in variational inference, diffusion model fixes $Q$ to a predefined Markov chain:

$$Q(y_t|y_{t-1}) = \mathcal{N}(\sqrt{1 - \beta_t} y_{t-1}, \beta_t I)$$

$$Q(y_1, \cdots, y_T|y_0) = \prod_{t=1}^{T} Q(y_t|y_{t-1}),$$

where $\beta_1, \cdots, \beta_T$ is a sequence of predefined scalars controlling the variance schedule. Since $Q$ is fixed, the last term $-\mathbb{E}_Q D_{\mathrm{KL}}(Q(y_T|y_0) \| P(y_T))$ in Equation (4) is a constant, and we only need to optimize

$$\mathbb{E}_Q \left[ \log P(y_0|y_1) - \sum_{t=2}^{T} D_{\mathrm{KL}}(Q(y_{t-1}|y_t, y_0) \| P(y_{t-1}|y_t)) \right]$$

$$= \mathbb{E}_{Q(y_1|y_0)} \log P(y_0|y_1) - \sum_{t=2}^{T} \mathbb{E}_{Q(y_t|y_0)} D_{\mathrm{KL}}(Q(y_{t-1}|y_t, y_0) \| P(y_{t-1}|y_t)). \quad (5)$$

Since in practice $T$ is usually set to a large number (this paper uses 1000), to efficiently compute the gradients, we use stochastic gradient descent and sample $t$ uniformly from $\{1, \cdots, T\}$ instead of computing the full sum $\sum_{t=2}^{T} D_{\mathrm{KL}}(Q(y_{t-1}|y_t, y_0)\|P(y_{t-1}|y_t))$ to estimate gradients. The key observation is that sampling from $Q(y_t|y_0)$ is efficient since $Q(y_t|y_0) = \int_{y_1, \cdots, y_{t-1}} Q(y_{1:t}|y_0)$ has an analytical form:

$$Q(y_t|y_0) = \mathcal{N}(\sqrt{\bar{\alpha}_t}y_0, \sqrt{1-\bar{\alpha}_t}I),$$

where $\bar{\alpha}_t = \prod_{s=1}^{t} \alpha_s$ and $\alpha_t = 1 - \beta_t$. This enables fast sampling of $y_t \sim Q(y_t|y_0)$ without sampling the intermediate steps $y_1, \cdots, y_{t-1}$:

$$\epsilon \sim \mathcal{N}(0, I)$$
$$y_t = \sqrt{\bar{\alpha}_t}y_0 + \sqrt{1-\bar{\alpha}_t}\epsilon.$$

We refer readers to Ho et al. (2020) for detailed derivations. Based on this observation, $Q(y_{t-1}|y_t, y_0) = \frac{Q(y_t|y_{t-1}, y_0)Q(y_{t-1}|y_0)}{Q(y_t|y_0)}$ is also a Gaussian:

$$Q(y_{t-1}|y_t, y_0) = \mathcal{N}(\frac{\sqrt{\bar{\alpha}_{t-1}}\beta_t}{1-\bar{\alpha}_t}y_0 + \frac{\sqrt{\alpha_t}(1-\bar{\alpha}_{t-1})}{1-\bar{\alpha}_t}y_t, \frac{(1-\bar{\alpha}_{t-1})}{1-\bar{\alpha}_t}\beta_t I)$$
$$= \mathcal{N}(\frac{1}{\sqrt{1-\beta_t}}(y_t - \frac{\beta_t}{\sqrt{1-\bar{\alpha}_t}}\epsilon), \frac{(1-\bar{\alpha}_{t-1})}{1-\bar{\alpha}_t}\beta_t I)$$

Since our goal is to minimize $\sum_{t=2}^{T} D_{\mathrm{KL}}(Q(y_{t-1}|y_t, y_0)\|P(y_{t-1}|y_t))$, to match the covariance matrix of $P(y_{t-1}|y_t)$ to that of $Q(y_{t-1}|y_t, y_0)$, we can simply set

$$\sigma_t^2 = \frac{(1-\bar{\alpha}_{t-1})}{1-\bar{\alpha}_t}\beta_t.$$

To match the mean, $\mu_\theta(y_t, t)$ should match

$$\frac{1}{\sqrt{\alpha_t}}(y_t - \frac{\beta_t}{\sqrt{1-\bar{\alpha}_t}}\epsilon).$$

Since $y_t$ is given as input to the model, Ho et al. (2020) propose to parameterize $\mu_\theta(y_t, t)$ in a similar form by parameterizing another function $\epsilon_\theta(y_t, t)$:

$$\mu_\theta(y_t, t) = \frac{1}{\sqrt{\alpha_t}}(y_t - \frac{\beta_t}{\sqrt{1-\bar{\alpha}_t}}\epsilon_\theta(y_t, t)).$$

With this parameterization the objective in Equation (5) can be written as (ignoring constants)

$$\sum_{t=1}^{T} \mathbb{E}_{Q(y_t|y_0)} \frac{\beta_t^2}{2\sigma_1^2 \alpha_t(1-\bar{\alpha}_t)} \|\epsilon_t - \epsilon_\theta(y_t, t)\|^2,$$

where $\epsilon_t = \frac{y_t - \sqrt{\bar{\alpha}_t}y_0}{\sqrt{1-\bar{\alpha}_t}}$. Ho et al. (2020) further proposes to ignore the weights $\frac{\beta_t^2}{2\sigma_1^2\alpha_t(1-\bar{\alpha}_t)}$, so the final objective becomes

$$\sum_{t=1}^{T} \mathbb{E}_{Q(y_t|y_0)} \|\frac{y_t - \sqrt{\bar{\alpha}_t}y_0}{\sqrt{1-\bar{\alpha}_t}} - \epsilon_\theta(y_t, t)\|^2. \tag{6}$$

## C  ALGORITHM OUTLINE

The training algorithm of scheduled sampling is shown in Algorithm 1, where the differences from the original DDPM training (shown in Algorithm 2) are highlighted in red. We refer readers to Ho et al. (2020) for the sampling algorithm, since our approach doesn't change sampling.

| **Algorithm 1** Scheduled Sampling | **Algorithm 2** No Scheduled Sampling |
|---|---|
| **Require:** $m \geq 0$ | **Require:** |
| 1: **repeat** | 1: **repeat** |
| 2: $\quad y_0 \sim$ data | 2: $\quad y_0 \sim$ data |
| 3: $\quad t \sim \text{Uniform}(\{1, \ldots, T-m\})$ | 3: $\quad t \sim \text{Uniform}(\{1, \ldots, T\})$ |
| 4: $\quad \epsilon \sim \mathcal{N}(0, I)$ | 4: $\quad \epsilon \sim \mathcal{N}(0, I)$ |
| 5: $\quad y_{t+m} \leftarrow \sqrt{\bar{\alpha}_{t+m}} y_0 + \sqrt{1 - \bar{\alpha}_{t+m}} \epsilon$ | 5: $\quad y_t \leftarrow \sqrt{\bar{\alpha}_t} y_0 + \sqrt{1 - \bar{\alpha}_t} \epsilon$ |
| 6: $\quad$ **for** $m' = m-1, m-2, \cdots, 0$ **do** | 6: |
| 7: $\qquad y_{t+m'} \sim P(y_{t+m'} \| y_{t+m'+1})$ | 7: |
| 8: $\quad$ **end for** | 8: |
| 9: $\quad$ Take gradient descent step on | 9: $\quad$ Take gradient descent step on |
| $\qquad \nabla_\theta \left\| \epsilon - \epsilon_\theta(y_t, t) \right\|^2$ | $\qquad \nabla_\theta \left\| \epsilon - \epsilon_\theta(y_t, t) \right\|^2$ |
| 10: **until** converged | 10: **until** converged |

Figure 10: The generation process of diffusion (left) versus diffusion+schedule sampling (right). The numbers on the y-axis are the number of diffusion steps ($t$). The ground truth LaTeX is `\tilde{Q}_{1}^{(L,I)} \tilde{Q}_{0}^{(L,I)} \neq 0 \,`

## D  MORE QUALITATIVE EXAMPLES OF INTERMEDIATE STEPS

More qualitative examples of the generative process comparing a model trained with scheduled sampling and a model trained without scheduled sampling can be found at Figure 10, Figure 11, and Figure 12.

## E  MORE PERTURBATION ANALYSIS

We expand our perturbation analysis in Figure 4 to more perturbation patterns. In addition to the deletion of symbols, we add the results for insertion and substitution. The distribution of the newly introduced symbols follows the empirical distribution of those symbols in the training data. We summarize the results in Table 3 and Figure 13.

## F  STABLE DIFFUSION EXPERIMENTAL DETAILS

We use the HuggingFace implementation of Stable Diffusion (Rombach et al., 2022). Stable Diffusion is originally trained on images with the size of $500 \times 500$. During inference, it generates

| 850 | $g_{00}=1,$ | |
| 750 | $g_{00}=1,$ | |
| 650 | $g_{00}=1,$ | $g_{ij}=-\left(\dfrac{22}{1+\vec{x}}\right)^2 a_{ij}$ |
| 600 | $g_{00}=1,$ | $g_{ij}=-\left(\left(\dfrac{2}{1-x^2}\right)^2\right)\delta_{ij}$ |
| 550 | $g_{00}=1,$ | $g_{ij}=-\left(\dfrac{2}{1-x^{2}}\right)^{4}\delta_{ij}$ |
| 500 | $g_{00}=1,$ | $g_{ij}=--\left(\dfrac{2}{1+\bar{x}^2}\right)^2\delta_{ij}$ |
| 450 | $g_{00}=1,$ | $g_{ij}=--\left(\dfrac{2}{1+\bar{x}^2}\right)^2\delta_{ij}$ |
| 350 | $g_{00}=1,$ | $g_{ij}=--\left(\dfrac{2}{+\bar{x}^2}\right)^2\delta_{ij}$ |
| 250 | $g_{00}=1,$ | $g_{ij}=--\left(\dfrac{2}{+\vec{x}'^2}\right)^2\delta_{ij}$ |
| 0 | $g_{00}=1,$ | $g_{ij}=--\left(\dfrac{2}{+\vec{x}'^2}\right)^2\delta_{ij}$ |

| 850 | $g_{00}=1,$ | |
| 750 | $g_{00}=1,$ | |
| 650 | $g_{00}=1,$ | $g_i=-\left(\dfrac{2}{1+\vec{x}^2}\right)^{2}a_{ij}$ |
| 600 | $g_{00}=1,$ | $g_{ij}=-\left(\dfrac{2}{1+\vec{x}^2}\right)^2 a_{ij}$ |
| 550 | $g_{00}=1,$ | $g_{ij}=-\left(\dfrac{2}{1+\vec{x}^2}\right)^2\delta_{ij}$ |
| 500 | $g_{00}=1,$ | $g_{ij}=-\left(\dfrac{2}{1+\vec{x}^2}\right)^2\delta_{ij}$ |
| 450 | $g_{00}=1,$ | $g_{ij}=-\left(\dfrac{2}{1+\vec{x}}\right)^2\delta_{ij}$ |
| 350 | $g_{00}=1,$ | $g_{ij}=-\left(\dfrac{2}{1+\vec{x}}\right)^2\delta_{ij}$ |
| 250 | $g_{00}=1,$ | $g_{ij}=-\left(\dfrac{2}{1+\vec{x}}\right)^2\delta_{ij}$ |
| 0 | $g_{00}=1,$ | $g_{ij}=-\left(\dfrac{2}{1+\vec{x}}\right)^2\delta_{ij}$ |

Figure 11: The generation process of diffusion (left) versus diffusion+schedule sampling (right). The numbers on the y-axis are the number of diffusion steps ($t$). The ground truth LaTeX is
`g_{00}=1,\qquad g_{ij}=-({\frac{2}{1+\vec{x}^{2}}})^{2}\delta_{ij}.`

| 850 | $\{\psi(\mathbf{x}),\psi^{\dagger}(\mathbf{y})\}=\delta(x-y)$ | $\{\psi(\mathbf{x}),\psi^{\dagger}(\mathbf{y})\}=\delta(x-y)$ |
| 750 | $\{\psi(\mathbf{x}),\psi^{\dagger}(\mathbf{y})\}=\delta(x-y)$ | $\{\psi(\mathbf{x}),\psi^{\dagger}(\mathbf{y})\}=\delta(x-y)$ |
| 650 | $\{\psi(\mathbf{x}),\psi^{\dagger}(\mathbf{y})\}=\delta(x-y)$ | $\{\psi(\mathbf{x}),\psi^{\dagger}(\mathbf{y})\}=\delta(x-y)$ |
| 600 | $\{\psi(\mathbf{x}),\psi(\mathbf{y})\}=\delta(x-y)$ | $\{\psi(\mathbf{x}),\psi^{\dagger}(\mathbf{y})\}=\delta(x-y)$ |
| 550 | $\{\psi(\mathbf{x}),\hat{\psi}(\mathbf{y}))\}=\delta(x-y)$ | $\{\psi(\mathbf{x}),\psi^{\dagger}(\mathbf{y})\}=\delta(x-y)$ |
| 500 | $\{\psi(\mathbf{x}),\hat{\psi}(\mathbf{y}))\}=\delta(x-y)$ | $\{\psi(\mathbf{x}),\psi(\mathbf{y})\}=\delta(x-y)$ |
| 450 | $\{\psi(\mathbf{x}),\psi(\mathbf{y}))\}=\delta(x-y)$ | $\{\psi(\mathbf{x}),\psi^{\dagger}(\mathbf{y})\}=\delta(x-y)$ |
| 350 | $\{\psi(\mathbf{x}),\psi(\mathbf{y})\}\}=\delta(x-y)$ | $\{\psi(\mathbf{x}),\psi^{\dagger}(\mathbf{y})\}=\delta(x-y)$ |
| 250 | $\{\psi(\mathbf{x}),\psi(\mathbf{y})\}\}=\delta(x-y)$ | $\{\psi(\mathbf{x}),\psi^{\dagger}(\mathbf{y})\}=\delta(x-y)$ |
| 0 | $\{\psi(\mathbf{x}),\psi(\mathbf{y})\}\}=\delta(x-y)$ | $\{\psi(\mathbf{x}),\psi^{\dagger}(\mathbf{y})\}=\delta(x-y)$ |

Figure 12: The generation process of diffusion (left) versus diffusion+schedule sampling (right). The numbers on the y-axis are the number of diffusion steps ($t$). The ground truth LaTeX is
`\{\psi({\bf x}),\psi^{\dagger}({\bf y})\}=\delta({\bf x-y}).`

| Approach | Math | | Simple Tables | | Music | | Molecules | |
|---|---|---|---|---|---|---|---|---|
| | DTW↓ | RMSE↓ | DTW↓ | RMSE↓ | DTW↓ | RMSE↓ | DTW↓ | RMSE↓ |
| **Gold Images** | | | | | | | | |
| -1 Symbol | 12.33 | 27.17 | 10.72 | 40.93 | 39.38 | 25.04 | 19.42 | 30.36 |
| -2 Symbols | 16.00 | 35.55 | 16.37 | 60.46 | 48.45 | 30.82 | 23.43 | 36.76 |
| -3 Symbols | 18.49 | 37.84 | 18.45 | 67.86 | 57.08 | 35.49 | 24.73 | 37.80 |
| -4 Symbols | 20.76 | 40.17 | 20.80 | 73.72 | 66.90 | 39.59 | 23.96 | 36.68 |
| -8 Symbols | 24.60 | 43.18 | 23.56 | 81.52 | 75.08 | 43.48 | 21.99 | 32.84 |
| -16 Symbols | 27.49 | 43.54 | 25.09 | 84.66 | 71.95 | 42.37 | 21.10 | 31.06 |
| +1 Symbol | 11.94 | 26.23 | 5.92 | 22.80 | 40.07 | 25.00 | 22.81 | 35.23 |
| +2 Symbols | 16.25 | 35.75 | 11.00 | 39.83 | 57.56 | 35.22 | 24.55 | 38.32 |
| +3 Symbols | 20.66 | 38.09 | 16.15 | 56.92 | 63.86 | 38.63 | 24.09 | 36.74 |
| +4 Symbols | 21.50 | 40.91 | 19.72 | 69.60 | 67.30 | 40.23 | 22.52 | 33.98 |
| +8 Symbols | 27.62 | 45.91 | 23.86 | 80.75 | 76.06 | 44.79 | 20.43 | 29.87 |
| +16 Symbols | 31.62 | 48.79 | 25.75 | 85.77 | 81.56 | 47.61 | 20.34 | 29.72 |
| $\Delta$1 Symbol | 16.55 | 31.42 | 11.20 | 41.31 | 52.88 | 32.20 | 15.25 | 24.27 |
| $\Delta$2 Symbols | 19.04 | 38.90 | 18.43 | 64.99 | 64.21 | 37.50 | 20.25 | 30.85 |
| $\Delta$3 Symbols | 23.42 | 41.58 | 20.96 | 72.71 | 67.73 | 39.60 | 21.11 | 31.27 |
| $\Delta$4 Symbols | 25.65 | 43.04 | 23.35 | 79.71 | 66.76 | 39.62 | 20.85 | 30.73 |
| $\Delta$8 Symbols | 28.61 | 45.89 | 25.57 | 86.93 | 75.37 | 43.97 | 20.34 | 29.72 |
| $\Delta$16 Symbols | 31.34 | 47.67 | 25.95 | 85.96 | 74.01 | 44.12 | 20.34 | 29.72 |
| **Scheduled Sampling** | 18.81 | 37.19 | 5.64 | 21.11 | 79.81 | 44.70 | 24.80 | 37.92 |

Table 3: Perturbation results. Perturbation noise is denoted by (-) deletion, (+) insertion, and ($\Delta$) substitution.

poor-quality images if we query significantly smaller-sized images. Therefore, for our Math domain experiments, we first generate images with the size of $320 \times 1600$ and then down-sample the generated images to $64 \times 320$. Figure 14 illustrates some of the images generated by Stable Diffusion.

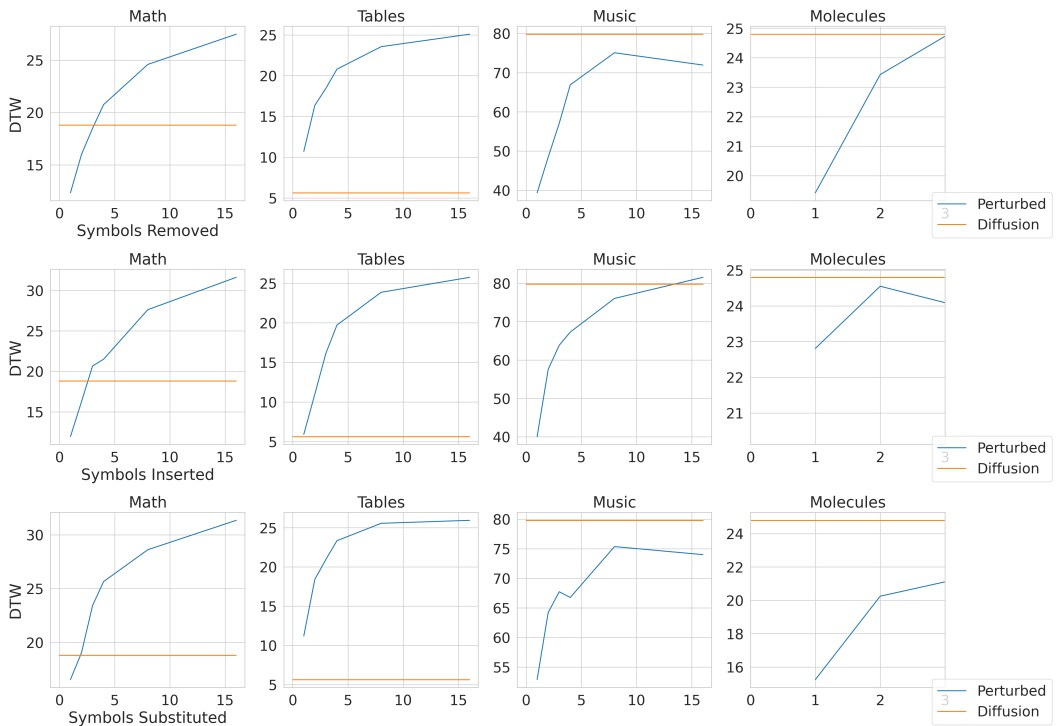

Figure 13: Perturbation results. Perturbation noise is denoted by (-) deletion, (+) insertion, and (Δ) substitution.

$$W^{(0)} \equiv \ln Z^{(0)} = W_\psi^{(0)} + W_A^{(0)},$$

$$\int dx\, f(x, y, t) = \int dx\, g(x, y, t).$$

$$D^\mu \frac{\delta f(A_\nu)}{\delta A_\mu} = D_\mu \partial^\mu (\partial_\nu A^\nu)$$

$$\mathcal{F}_{\{q,r,t\}} \left( \xi_{\{q,r,t\}} \right) := -$$

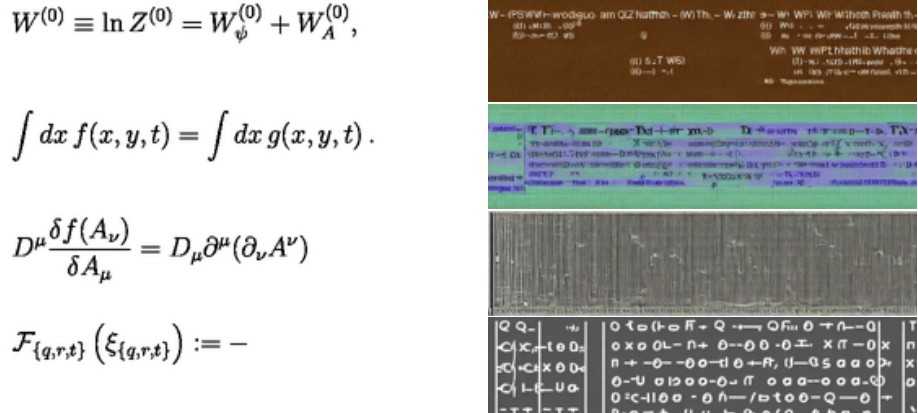

Figure 14: Qualitative results from Stable Diffusion (Rombach et al., 2022) in the Math domain. Left column: ground truth images. Right column: generations from Stable Diffusion.

