# OpenReview forum: "Markup-to-Image Diffusion Models with Scheduled Sampling"
_ICLR.cc/2023/Conference — ICLR 2023 poster_

### Official Review · Reviewer_LE3F · 2022-10-21

**Confidence:** 4
**Correctness:** 3
**Technical Novelty And Significance:** 2
**Empirical Novelty And Significance:** 4
**Recommendation:** 6

**Clarity, Quality, Novelty And Reproducibility:**

Overall, the paper is easy to follow although there were some typos. I find there is originality and novelty in the work but the discussion and the content of the work could go deeper as mentioned above. More meaningful experiments could be done to assess the quality of the latest image generation models.

**Strength And Weaknesses:**

Strengths:

There is a value to introduce the markup-to-image generation task for image generation models. It can assess the aspects of the models that cannot be assessed by the existing tasks and there is a potential that the new task brings fundamental improvements to the models.

It is reasonable to use scheduled sampling for diffusion-based models and the effectiveness is confirmed by the experiments.

Weaknesses:

It more or less just applied a diffusion-based image generation model to the markup-to-image generation task with a technique inspired by a known technique. We can see the results by the latest image generation algorithm on the task, but it is not clear what scientific value it has to show the results by the models optimized for other tasks. It could have a deeper discussion on why the new task is needed and why the existing method does not work well yet, and a scientific contribution based on the deeper insight.

Probably, the experiments on assessing the absolute quality of the model are not enough. The "Perturbed" results in Figure 4 seem to have been generated by removing symbols for the amount in the x-axis. This can only tell how the evaluation metric is affected by random deletion errors. I do not feel it tells much to compare the simulated results with the ones by the diffusion-based models. The readers do not know how critical it is to miss a symbol to a particular renderer. How about insertion and substitution errors?

**Summary Of The Paper:**

This paper proposes markup-to-image generation as a task to measure the quality of image generation models, and examines the quality of the images generated by diffusion-based models for the task. They considered four domains, math, table, sheet music, molecules. As an improvement to the diffusion-based models for the task, they propose to use scheduled sampling. The experiments confirm that it improves the performance. The code and data are open-sourced.

**Summary Of The Review:**

The introduction of the new task seems valuable and it could potentially have a huge impact by resulting in a number of following studies. However, the discussion and the content in the paper are rather shallow and the quality of the paper could be further improved. I do not see any concerns and fine to see it accepted, but it is not a very strong support.

---

> ### Author Response · Authors · 2022-11-19
> **Response to Reviewer LE3F**
>
> Thank you for your comments!
>
> ### Re: the scientific value of the task?
> Our proposed task is useful for objectively diagnosing behaviors of text-to-image generation models. Through the proposed task, we analyzed the state-of-the-art diffusion text-to-image generation models, and we found that 1) diffusion models have a frequent failure mode of repeating symbols during image generation and 2) they cannot correct their own mistakes during the generation process. This is a new finding to our knowledge.
>
> In addition, our proposed task addresses the challenges of objectively comparing image generation models by enabling the objective evaluation of models. Our proposed task has only one ground truth reference for each input, while for many existing text-to-image benchmarks, there exist many different generations that are all valid, making evaluation challenging.
>
> Finally, we believe interesting future work is to combine natural language and structured language in image generation. For example, combining scene descriptions in a document layout, or aligned images in a presentation slide. This is not explored in this work, but is an interesting potential area of exploration.
>
> ### Re: just applied known model with a known technique to a new task.
> We identified the common failure mode of repeating symbols problem for image generation of diffusion models and hypothesize this is similar to exposure bias problems. When we applied the known solution to the exposure bias problems (i.e., scheduled sampling), we saw a consistent improvement in image generation across different domains, with a slight change in the training algorithm (Appendix C shows the change in training procedure compared to vanilla diffusion training).
>
> To our knowledge, none of the previous works ever relate exposure bias problems to image generation, and we are the first to show the improvement in image generation by applying the technique to remedy exposure bias problems.
>
> ### Re: how about insertion and substitution errors?
> We have added the results of insertion and substitution in Appendix E.

---

### Official Review · Reviewer_aUAh · 2022-10-22

**Confidence:** 4
**Correctness:** 4
**Technical Novelty And Significance:** 2
**Empirical Novelty And Significance:** 3
**Recommendation:** 6

**Clarity, Quality, Novelty And Reproducibility:**

Clarity: High.

Quality: In general the quality on introducing the problem, challenge, and solution is high. The experiments could probably be better designed: especially the evaluation metrics, and if the evaluation metrics are not to be trusted or revealing, more analysis on the patten of failure cases and improvments would be helpful.

Novelty: The chosen methods (diffusion, scheduled sampling algorithm) are existing works, but the perspective of applying the methods to deterministic markup-to-image problem is novel.

Reproducibility: High.

**Strength And Weaknesses:**

Strength
1. This paper explores a unique generation problem with diffusion model where the mapping between input and generation target is deterministic, this provides a new angle for analyzing and evaluating generative models.
2.  Authors noticed that standard diffusion training process for the markup-to-image problem often generate duplicate or misplaced symbols similar to widely studied exposure bias, so they adapt scheduled sampling by training diffusion model with its own generations and showed improved evaluation metrics on four small-scaled datasets: Math, Tables, Music and Molecules.


Weaknesses
1. Diffusion model has been proven effective in distribution learning, if the learning obejctive is a deterministic mapping, would regression based methods be more appropriate ?
2. Comparisons are only made among authors' designs, are there previous works to compare with ?
3. The authors already hinted varying performance on different domains, I would like to see more analysis on it.

**Summary Of The Paper:**

This paper experiments on a deterministic generation (renderining) markup-to-image problem with diffusion model. They show that standard diffusion training process suffers exposure bias and using scheduled sampling algorithm is effective in fixing generation issues.

**Summary Of The Review:**

The  paper is clearly written. Treating markup-to-image as a generative process is a debatablly meaningful but novel point, especially when considering evaluating generative models. The authors use diffusion model as the generative model and adapt scheduled sampling to improve generation results. The methods are evaluated on four datasets with limited copmarison to other works if exist. The analysis of results is not totally convincing though, especially when the evaluation metrics are not convincing enough, more analysis on qualitative results would be helpful.

---

> ### Author Response · Authors · 2022-11-19
> **Response to Reviewer aUAh**
>
> Thank you for your comments!
>
> ### Re: would regression-based methods be more appropriate for markup-to-image tasks?
> We propose markup-to-image tasks for diagnosing the behavior of state-of-the-art text-to-image generation models. The deterministic nature of our task–-having only a single ground truth-–makes it possible to analyze the behavior of text-to-image generation models objectively. Therefore, diffusion models are appropriate for our study, given their dominant performance in text-to-image tasks. (We agree that diffusion models target distribution learning, but we do not think that precludes their use on simpler or even deterministic distributions as shown by the results here.)
>
> ### Re: are there previous works to compare with?
> Using the markup-to-image task, we show the issue of diffusion models being unable to correct their own mistakes, which to our knowledge, is a new finding. Our baseline implementation is based on Huggingface diffusers (forked on Sep 7 2022), which is one of the best available implementations of diffusion models, and we believe that showing improvement over this baseline is meaningful, especially considering that we only made a small change to the training procedure (see Appendix C, Algorithm 1 vs Algorithm 2).
>
> Pretrained diffusion models (e.g., Stable Diffusion) perform very poorly in the markup-to-image task. In Table 2, we added the evaluation of pre-trained Stable Diffusion (Rombach et al., 2022) for the Math dataset, and we provide qualitative results in Figure 14 (Appendix F). Our results suggest that the pre-trained Stable Diffusion model performs extremely poorly.
>
> ### Re: varying performance on different domains.
> We hypothesize that the varying performance on different domains is mainly due to the varying inherent complexity of the task in each domain. For example, the performance on the Table dataset is the best since the dataset is simple: just generating several symbols and a few enclosing boxes. On the other hand, the performance on the Music dataset is the worst since the task involves generating a long chain of notes which has strong dependencies on the preceding notes.

---

### Official Review · Reviewer_uJv3 · 2022-10-22

**Confidence:** 4
**Correctness:** 3
**Technical Novelty And Significance:** 3
**Empirical Novelty And Significance:** 3
**Recommendation:** 8

**Clarity, Quality, Novelty And Reproducibility:**

Overall, I found the paper to be mostly clear and of high quality. While the key ideas of the main technical innovation (i.e., scheduled sampling) exist in some form in the literature, I believe the application to diffusion models is novel, as is the task of markup-to-image generation to some extent. I don't forsee any problems with reproducibility.

I have a few additional suggestions for improving the clarity of the manuscript:
- Fig. 2 is showing the steps of the reverse diffusion, but the convention is that the steps number the forward diffusion process, i.e., step 0 is the clean image and the image converges to Gaussian noise with increasing steps. I think the opposite numbering convention is used here, which conflicts with the notation elsewhere.

- Apply consisting bolding in Table 2.

- Implementation details: uses M=1 but I think the parameter is given as "m" and not "M" previously in the manuscript.

- The paper should describe what is meant by a "gold" image, as this is not a standard term in my experience. Without this, I could not follow the perturbation analysis. I also didn't understand how the symbols are removed in the perturbation analysis. Are they removed from the input text conditioning?



**Strength And Weaknesses:**

The paper is mostly well-written and I found the description of the method to be clear. The paper makes a convincing case for the markup-to-image task, and I think this direction has potential for impact in the community. Moreover, the evaluation seems thorough, and while there is still room for improvment in the results, the manuscript and method are of high quality.

While I think the paper is fairly strong, I do have some reservations about the evaluation and clarity of the writing in a few sections as I detail below.
- The main quantitative results shown in Table 2 do not necessarily show the most convincing argument for the benefits of scheduled sampling. Some metrics are very close between the baselines, and scheduled sampling seems to worsen performance in other metrics (e.g., RASE). The paper also seems to be missing an analysis of how the performance changes with choosing m>1 (the number of scheduled sampling steps to use). Since I perceive the scheduled sampling to be the main technical contribution, I was expecting a more thorough analysis on this point.
- I see only one qualitative result that shows the effect of including scheduled sampling (Fig. 2). I think including more such results could help make a more convincing argument for using this procedure.
- I couldn't follow the perturbation analysis section. The paper repeatedly refers to a "gold" image without explaining what this is.


**Summary Of The Paper:**

The paper proposes the task of markup-to-image generation and adapts a procedure of scheduled sampling to improve diffusion model performance. Here, the objective is to synthesize images from latex, musical notation, or HTML snippets, or any markup language that defines an output image in a deterministic fashion. In this context, scheduled sampling is applied to incorporate intermediate outputs of the diffusion model during training. Empirically, this results in some ability for the model to self-correct when it begins to generate erroneous outputs. The method is evaluated on a few different proposed datasets (HTML, music notation, molecule diagrams, latex) with an array of ablation studies. Overall the results show convincing performance, and this research direction may be significant for advances in markup compilers that are robust to typos or which incorporate generative capabilities.

**Summary Of The Review:**

Overall the paper is well-written and contributes an interesting new technical idea for improving diffusion models as well as a new and potentially impactful task of markup-to-image generation. While there are areas where the clarity of the manuscript and the evaluation could be improved, I think the paper meets the bar for acceptance.

---

> ### Author Response · Authors · 2022-11-19
> **Response to Reviewer uJv3**
>
> Thank you for your comments!
>
> ### Re: scheduled sampling shows little improvement for some metrics in Table 2.
> Training with scheduled sampling consistently gives benefits on all datasets, with particularly strong gains on three out of four datasets (Math, Table Layouts, and Sheet Music) on pixel level metrics. Considering that scheduled sampling only makes a small change to the training algorithm and does not change other hyperparameters (see Appendix C Algorithm 1 vs Algorithm 2), we think the evidence is strong that scheduled sampling improves performance.
>
>
> ### Re: missing an analysis of how the performance changes when m>1.
> The results of using m>1 on the math dataset are shown in the below table. For this experiment, we use the same setting of linearly increasing the probability of applying scheduled sampling  from 0 (0 epoch) to 0.5 (at 100 epochs). If scheduled sampling is applied, we uniformly sample an m value from 1 to the maximum value of m. All models are trained for 20 epochs due to time constraints.
> | max m    | DTW      | RMSE|
> | ----------- | ----------- |----------|
> | 0            |   24.94   | 42.89  |
> | 1            |   24.34   | 42.11  |
> | 2            |   23.89   | 41.35  |
> | 4           |  24.50    | 42.63  |
> | 8           |   23.66   | 41.14  |
>
> From this table we can see that using m>1 can potentially further improve performance, although using larger m values will lead to slower training. We will add extra experiments on how m affects accuracy and training speed in our next version.
>
> ### Re: more qualitative results.
> We have included more qualitative results showing the effect of using scheduled sampling in Appendix D.
>
> ### Re: perturbation analysis unclear.
> To obtain a perturbed image, we take the input markup, uniformly at random remove n symbols and use the compiler to compile the perturbed input. We refer to the compiled image of the original markup as the "gold" image. We have changed the wording of this section to make it more clear.
>
> ### Re: styling issues.
> Thanks for catching these issues.
> - In Figure 2 and the newly added qualitative results, we have changed the y axis to show t instead of T-t, such that step 0 is the clean image.
> - We have changed M to m in Section 4.3
> - Regarding consistent bolding in Table 2, do you mean we should bold other columns in Table 2 in addition to the first two columns? We have added bolding to other columns in the revised version. We want to note that our main evaluation metrics are still the first two columns (pixel-level metrics), since other metrics do not separate different approaches well. For example, the lowest CLIP score is 0.95, even though the generations in that baseline look terrible qualitatively.

---

### Official Review · Reviewer_NDpz · 2022-10-25

**Confidence:** 3
**Correctness:** 3
**Technical Novelty And Significance:** 2
**Empirical Novelty And Significance:** 2
**Recommendation:** 3

**Clarity, Quality, Novelty And Reproducibility:**

I think there're still plenty of aspects that the authors need to address before getting accepted.

**Strength And Weaknesses:**

Strength:
+ The proposed algorithm seems to be effective.

Weakness:
- The algorithm is not well explained. I believe that readers require a lot of background in diffusion model to understand the paper content. Some notions are proposed without any explanation or reference, which makes the paper even harder to follow. To name just a few, starting from Eqn. 1 (Sec.3), the paper talks about the Markov chain Q and its sampling. However, what is Q and how Q is related to diffusion model is not discussed. How to derive Eqn.1, and Eqn. 2 are also unknown. I would like to suggest the authors either briefly introduce the pipeline and refer readers to a more detailed deduction, or carefully explain the terms to avoid ambiguity.
- The overall network pipeline is unclear. I think including a pipeline figure and an algorithm flow tables for both training and sampling procedures are necessary.
- Lacking essential experiments. The paper only show results from their own and don't include any comparison results with state-of-the-art (SOTA) text-to-image generative models. Considering the large amount of SOTAs, I believe that a comprehensive comparison result is needed.

**Summary Of The Paper:**

This paper proposed a novel scheduled sampling algorithm to guide the training of diffusion models for markup-to-image generation. Unlike the traditional diffusion models, which sample the training data from the Markov chain quantified Gaussian distribution Q(yt|y0), the paper take the m earlier predictions into consideration and thus can address the exposure bias issues in the image generative models to some extent.

**Summary Of The Review:**

In general, the paper proposed a novel sampling strategy for diffusion models to handle exposure issues. Even though the paper provided some reasonable results, lacking essential comparison experiments and poor paper organization make me hesitate to give it a high rank.

---

> ### Author Response · Authors · 2022-11-19
> **Response to Reviewer NDpz**
>
> Thank you for your comments.
>
> ### Re: the algorithm is not well explained.
>
> We have added the following additional details in Appendix B due to space. The base model is complex, so we focused in the body on the scheduled sampling aspect.
>
> - Regarding the meaning of Q, we added an explanation “To learn the model, we use variational inference and maximize the evidence lower bound (ELBO) (Bishop & Nasrabadi, 2006). We introduce an auxiliary Markov chain $Q(y_1, \cdots, y_T|y_0) = \prod_{t=1}^TQ(y_{t} | y_{t-1})$ as the approximate posterior distribution to compute the ELBO:” (page 16)
> - Regarding the derivation of Eqn. 1, we have expanded the intermediate steps in Eqn. 4 (page 16)
> - Regarding the derivation of Eqn. 2, we have expanded the intermediate steps on page 17.
>
> ### Re: the overall network pipeline is unclear.
> We have added following contents in Appendix C to address the reviewer’s concern:
> - We have added an illustration of the training algorithm in comparison with the original DDPM training algorithm.
> - We also referred readers to the original DDPM paper for the sampling algorithm in Appendix C since our approach only changes training but not sampling.
>
> ### Re: lacking essential experiments comparing to SOTA.
> - The reviewer may mean pretrained state-of-the-art text-to-image diffusion models. In Table 2, we added the evaluation of pre-trained Stable Diffusion (Rombach et al., 2022) for the Math dataset, and we provide qualitative results in Figure 14 (Appendix F). Our results suggest that the pre-trained model performs extremely poorly though. It does not have a zero-shot transferability to the markup-to-image tasks since they are trained mainly on natural language texts.
>
> - The reviewer may mean state-of-the-art architectures. In that case we note our baseline implementation is based on Huggingface diffusers (forked on Sep 7 2022), which is one of the best available implementations of diffusion models to our knowledge.
>
> In either case clarity into this comment would be helpful. We also note that the proposed scheduled sampling algorithm intends to solve the exposure bias problem during DDPM training. The improvements brought by our training adaptation should be orthogonal to the improvements from other sources (better models, better sampling, etc.), even if there are other types of diffusion models.

---

### Decision · Program_Chairs · 2023-01-20

**Decision:**

Accept: poster

**Justification For Why Not Higher Score:**

The paper presentation and experiment results are a bit lacking.

**Justification For Why Not Lower Score:**

The idea is novel enough and the evaluation is sufficient.

**Metareview: Summary, Strengths And Weaknesses:**

The paper proposes using scheduling sampling when training a diffusion model. It starts by introducing the markup-to-image generation task, which is a task of generating an image from the markup description of the image. This task has a one-to-one mapping between the input and the output. It is particularly suitable for evaluation. In this task, they compare training diffusion models with scheduling sampling or not. Experiment results suggest that with scheduling sampling, the trained diffusion models produce higher-quality images.

The paper receives four reviewers. Three reviewers consider the paper above the bar, citing the task novel, the idea of using scheduling sampling well-motivated, and the presented results convincing. One reviewer rates the paper below the bar. It considers the presented algorithm not well-explained and the presented results lacking those from state-of-the-art methods.

After consolidating the paper, review, and rebuttal, the AC decides to side with the majority. The AC agrees the presented task is interesting and the proposed method well-motivated. While the AC agrees that the algorithm can be better explained and the authors could have included more stronger baselines, the AC would not hold them against acceptance of the paper. The AC believes the paper has enough value to be published as an ICLR paper.

**Note From Pc:**

if the above contains the word "oral" or "spotlight" please see: "oral" presentation means -> notable-top-5% and "spotlight" means -> notable-top-25%. As stated in our emails, we are disassociating presentation type from AC recommendations

**Summary Of Ac-Reviewer Meeting:**

N/A